# Competing correlated states and abundant orbital magnetism in twisted monolayer-bilayer graphene

Minhao He [1], Ya-Hui Zhang[2], Yuhao Li[1], Zaiyao Fei [1], Kenji Watanabe [3], Takashi Taniguchi [4], Xiaodong Xu [1,5✉] & Matthew Yankowitz [1,5✉]

Flat band moiré superlattices have recently emerged as unique platforms for investigating the interplay between strong electronic correlations, nontrivial band topology, and multiple iso-spin 'flavor' symmetries. Twisted monolayer-bilayer graphene (tMBG) is an especially rich system owing to its low crystal symmetry and the tunability of its bandwidth and topology with an external electric field. Here, we find that orbital magnetism is abundant within the correlated phase diagram of tMBG, giving rise to the anomalous Hall effect in correlated metallic states nearby most odd integer fillings of the flat conduction band, as well as correlated Chern insulator states stabilized in an external magnetic field. The behavior of the states at zero field appears to be inconsistent with simple spin and valley polarization for the specific range of twist angles we investigate, and instead may plausibly result from an intervalley coherent (IVC) state with an order parameter that breaks time reversal symmetry. The application of a magnetic field further tunes the competition between correlated states, in some cases driving first-order topological phase transitions. Our results underscore the rich interplay between closely competing correlated ground states in tMBG, with possible implications for probing exotic IVC ordering.

[1] Department of Physics, University of Washington, Seattle, WA, USA. [2] Department of Physics, Harvard University, Cambridge, MA, USA. [3] Research Center for Functional Materials, National Institute for Materials Science, Tsukuba, Japan. [4] International Center for Materials Nanoarchitectonics, National Institute for Materials Science, Tsukuba, Japan. [5] Department of Materials Science and Engineering, University of Washington, Seattle, WA, USA. ✉email: xuxd@uw.edu; myank@uw.edu

In twisted graphene heterostructures with flat electronic bands, Coulomb interactions can spontaneously lift the degeneracy between spin, orbital, and lattice flavor symmetries[1–6]. In the simplest case, the many-body ground state is completely polarized into a subset of these isospin flavors. However, a much wider family of correlated ground states is also possible, including various density-wave orders[7] and exotic quantum spin liquid states[8]. Among these, theoretical calculations often find inter-valley coherent (IVC) states to be competitive with Ising-like valley polarized (VP) states at zero magnetic field[9–11]. In magic-angle twisted bilayer graphene (tBLG), certain IVC states have been proposed as the parent ground state out of which super-conductivity emerges[10,12,13], although direct experimental identification of the ground state order is challenging. Twisted monolayer-bilayer graphene (tMBG) features lower crystal symmetry, and consequentially the lattice degeneracy is strongly lifted at the single-particle level in an external displacement field, $D$. The lowest moiré conduction band has four remaining degenerate copies corresponding to spin and valley, and is flat enough to host correlated states over a small range of twist angles[14–16]. The bandwidth, $W$, and valley Chern number, $C_v$, additionally depend on the orientation of $D$ (Supplementary Fig. 1)[17–19], which can polarize charge carriers more strongly towards either the mono-layer or Bernal-stacked bilayer graphene sheet[14]. The high tun-ability of the bands with the combination of twist angle, doping, $D$, and magnetic field makes tMBG an attractive platform for investigating the nature of closely competing correlated and topological ground states.

## Results

**Anomalous Hall effect in metallic states of tMBG**. Here, we report electrical transport measurements from three tMBG sam-ples over a tight range of twist angles, $1.13° \leq \theta \leq 1.19°$. Notably, our studied range of twist angles falls intermediate to recent prior reports[14–16], enabling a more complete understanding of the evolution of correlated states in tMBG with $\theta$. We primarily focus our attention on two devices with twist angles of $\theta = 1.13°$ (device D1) and $1.19°$ (device D2). Figures 1a, b show maps of the longitudinal resistivity, $\rho_{xx}$, in device D1 for $D > 0$ and $D < 0$, respectively. The maps are primarily confined to the flat con-duction band ($0 \leq v \leq 4$, where $v$ is the band filling factor as defined in Methods) and large $|D|$, for which correlated states are observed at low temperatures. We assume a convention in which $D > 0$ corresponds to the electric field pointing from the mono-layer to the bilayer graphene. We observe robust insulating states at $v = 0$ and 4 for both signs of $D$, as anticipated from calculations of the single-particle band structure of tMBG. We additionally see well-developed correlated insulating states at $v = 1$ and 2 for $D > 0$ (Fig. 1a), as well as correlated metallic states at $v = 1$, 2, and 3 for $D < 0$ (Fig. 1b). Here, we define insulating states as those exhi-biting increasing $\rho_{xx}$ as the temperature is lowered, whereas metallic states behave oppositely (see Supplementary Figs. 7a, b and 3c, d for the temperature dependence of $\rho_{xx}$).

This correlated phase diagram connects smoothly to prior measurements of devices with slightly different twist angles. For $D > 0$, devices over a wide range of twist angles manifest a robust correlated insulating state at $v = 2$[14–16]. However, the states at $v = 1$ and 3 are absent in devices with slightly smaller twist angles (although appear to re-emerge in a device with even smaller twist angle, $\theta = 0.89°$[14], subsequent to an anticipated topological transition in the band from $C_v = 2$ to 1[18]). In contrast, both of these states are seen in devices with slightly larger twist angles[15,16], before eventually disappearing again in devices with $\theta \gtrsim 1.4°$[14–16]. The correlated states at $D < 0$ are less sensitive to twist angle, with resistive bumps observed at all integer $v$ in

devices with $1.05° \lesssim \theta \lesssim 1.4°$[14–16]. The absence of robust correlated insulating states is likely a consequence of the larger bandwidth compared with the $D > 0$ bands[17–19]. Supplementary Note 4 and Supplementary Table 1 provide a detailed summary of the correlated states observed in our three devices, as well as those previously reported in refs. [14–16].

In all three of our devices, we observe an anomalous Hall effect (AHE) within the "halo" region associated with the symmetry-broken state at $v = 1$ for $D > 0$ (see Fig. 1c, d and Supplementary Figs. 4 and 6). We see hysteretic behavior in both $\rho_{xx}$ and $\rho_{yx}$ as $B$ is swept back and forth at fixed $v$ and $D$ (Fig. 1c, d for device D1). Because spin-orbit coupling is extremely weak in graphene, spin-ordered magnetism is not anticipated to result in an AHE in tMBG. Instead, the AHE with hysteretic behavior is very likely tied to orbital magnetism[14,15,20–23]. Figure 1e, f show maps of the AHE versus $v$ at two different fixed $D > 0$, acquired by taking the difference of $\rho_{yx}$ between the two field sweeping directions, $\Delta\rho_{yx} = \rho_{yx}^{B\downarrow} - \rho_{yx}^{B\uparrow}$. Precisely at $v = 1$ in the map acquired at $D = 0.47$ V/nm (Fig. 1e), we measure a large $\rho_{xx}$ of tens of kiloohms and a rapidly oscillating $\Delta\rho_{yx}$ consistent with the behavior of a trivial insulating state, rather than a quantum anomalous Hall (QAH) state (see also Supplementary Fig. 8a, b). This contrasts prior reports in devices with $\theta = 1.25°$, in which a nearly quantized AHE in $\rho_{yx}$ and a small $\rho_{xx}$ are observed precisely at $v = 1$[15]. For $D < 0$, our measurements additionally reveal large pockets of AHE near $v = 1$ and 3 (Fig. 1g–j), which have previously not been reported.

**Ground state ordering of the $v = 1$ state**. The AHE has been observed previously in a number of graphene-based moiré plat-forms (tBLG[20,21], aligned ABC trilayer graphene on boron nitride[22], and tMBG[14,15]), with a tendency towards quantization precisely at odd integer $v$ (1 and/or 3). In the simplest case, a QAH state in tMBG is expected to arise upon polarization into a single spin- and valley-polarized band precisely at odd integer $v$ owing to the finite $C_v = 2$ of the conduction band. Quantization is lost upon doping with electrons or holes, however, an AHE can still persist owing to the large Berry curvature at the band extrema. Upon applying a magnetic field, the trajectory of the state drifts in the $n-B$ phase space as described by the Středa formula[24], $C = (h/e)(\partial n/\partial B)$, with slope equal to the Chern number, $C$, of the state ($h$ is Planck's constant, $e$ is the charge of the electron, and $n$ is the charge carrier density). All of these features have been observed previously in tMBG devices for $D > 0$ at a slightly larger twist angle ($\theta = 1.25°$) with $C = 2$[15]. Our results in devices with slightly smaller twist angles contrast these expectations, however, exhibiting neither signatures of a QAH state precisely at odd integer $v$, nor the anticipated finite sloping of these states in a weak magnetic field. Figure 2a shows the low-field Landau fan diagram at $D > 0$ in device D1, along with a cut of $\rho_{xx}$ versus $B$ taken precisely at $v = 1$ in the panel to the right. We find that the insulating state at $v = 1$ projects vertically and is suppressed with small $B$, before eventually re-emerging at larger $B$. Although in principle a disordered network of VP states with a nearly equal mixture of valley K and K' domains may localize to form a trivial insulator, the application of a weak $B$ should rapidly align the domains and form a Chern insulator state. Our obser-vations are inconsistent with this scenario, suggesting that the $B = 0$ ground state is not a VP state.

Figure 2b shows a similar measurement for $D < 0$ in device D2. This device is notable in that it exhibits the only known instance of insulating temperature dependence in a $D < 0$ correlated state, in this case arising at $v = 1$ (see temperature dependence in Supplementary Fig. 7c). Similar to the case of the $v = 1$ state in device D1 at $D > 0$, we find that $\rho_{xx}$ projects vertically, first

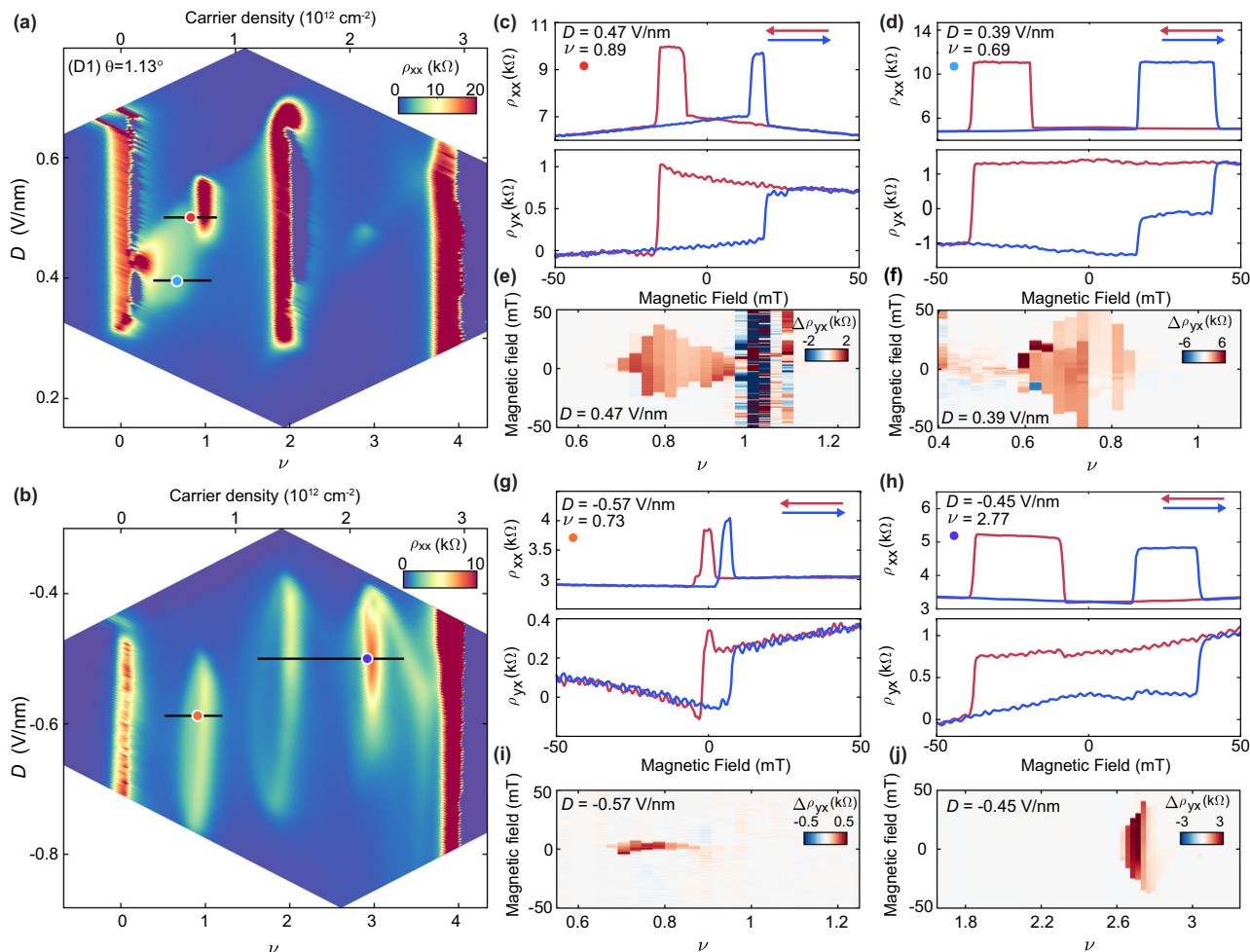

**Fig. 1 AHE in metallic states of tMBG. a, b** Longitudinal resistivity, $\rho_{xx}$, of device D1 ($\theta = 1.13°$) for $D > 0$ (**a**) and $D < 0$ (**b**). $\rho_{xx}$ is symmetrized at $|B| = 0.5$ T in order to suppress any magnetic hysteresis effects but looks very similar at $B = 0$ (Supplementary Fig. 3a). **c, d** $\rho_{xx}$ (top) and $\rho_{yx}$ (bottom) acquired as $B$ is swept back and forth at $\nu$ and $D$ indicated by the labels, and by the red and blue markers in (**a**), respectively. **e, f** Hysteresis loop height, $\Delta\rho_{yx}$, as a function of doping at $D = 0.47$ V/nm (**e**) and $D = 0.39$ V/nm (**f**), as indicated by the black lines in (**a**). Rapidly oscillating red and blue points near $\nu = 1$ in (**e**) arise due to the correlated trivial insulating state rather than magnetic ordering (see Supplementary Fig. 8a, b). **g, h** $\rho_{xx}$ (top) and $\rho_{yx}$ (bottom) acquired as $B$ is swept back and forth at $\nu$ and $D$ indicated by the labels, and by the orange and purple markers in (**b**), respectively. **i, j** $\Delta\rho_{yx}$ as a function of doping at $D = -0.57$ V/nm (**i**) and $D = -0.45$ V/nm (**j**), as indicated by the black lines in (**b**). We note that small oscillation features in $\rho_{yx}$ curves are noise, arising due to the low excitation current (1 nA) used in the measurements. $\rho_{yx}$ is not antisymmetrized and is offset from zero as a result of mixing with $\rho_{xx}$. All data are acquired at $T = 0.3$ K.

becoming less resistive with $B$ before eventually growing at high field. Both of these insulating states at $B = 0$ are accompanied by an AHE in a small region of $\nu < 1$, approximately corresponding to the regions of sharply enhanced resistivity in Fig. 2a, b. This is consistent with our understanding that the AHE is associated with the symmetry-broken state at $\nu = 1$, which persists well away from integer filling. Similar to the case of the $\nu = 1$ state for $D > 0$, we find that the AHE is strongest for $\nu \sim 0.9$ but vanishes at $\nu = 1$, where a trivial insulating state emerges instead (see Supplementary Fig. 4e, f). Similarly, this state is not consistent with full valley polarization.

A number of correlated ground states have previously been proposed for twisted double bilayer graphene (tDBG)[25] (see Methods), and are candidate ground states for tMBG as well. Among those that result in an insulating state at odd integer $\nu$, none naturally explain our observation of an AHE upon doping, as this requires a highly unusual combination of broken time-reversal symmetry (TRS), finite Berry curvature, but a net $C = 0$. In order to gain more insight, we start with a fully spin- and VP ground state at $\nu = 1$ and determine its stability by calculating the

dispersion of the collective spin and valley wave excitation (see Methods and Supplementary Note 1–3 for full details). The negative energy of this excitation is a signature of the instability of the spin-VP state. Figure 2c shows the calculated valley magnon energy for various values of the dielectric constant, $\epsilon$, which acts as an effective tuning parameter for the ratio $U/W$, where $U$ is the Coulomb interaction strength. We define a valley magnon as a spin-singlet valley-flip exciton (i.e., an exciton comprising two particles from opposite valleys). Condensation into an IVC order is favored when the valley magnon energy becomes negative at any point within the first Brillouin zone. We find that the VP state is stable at small $\epsilon$ (i.e., large $U/W$), however, there is a first-order phase transition into an IVC order above a modest critical $\epsilon$ (i.e., intermediate $U/W$). Figure 2d shows schematic representations of the VP and IVC states on the Bloch sphere.

Generically, the IVC state will retain TRS since it is a superposition of states at valleys K and K', trivializing the overall Chern number. However, our analysis of the IVC order parameter reveals that it prefers to carry finite angular momentum owing to the non-zero $C_\nu$ of the constituent bands,

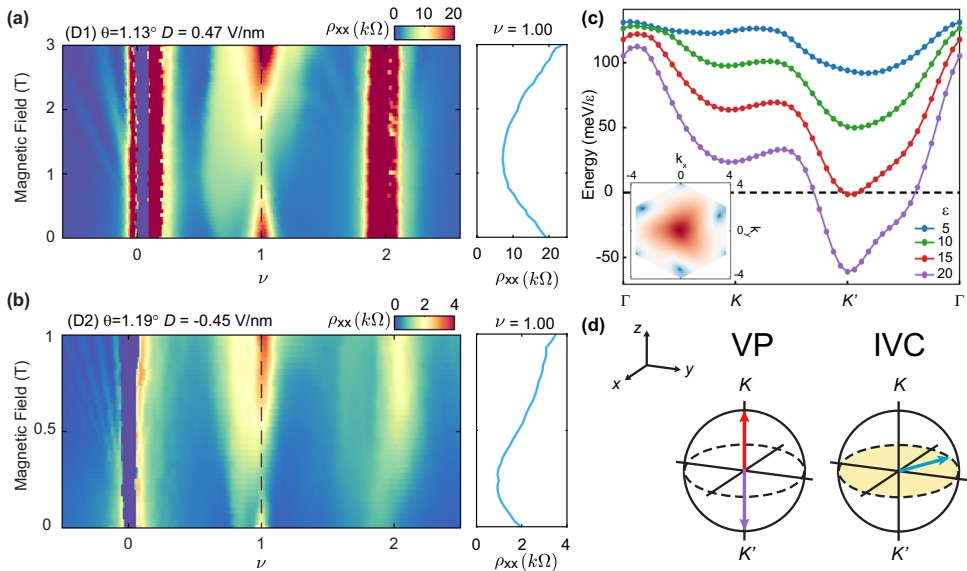

**Fig. 2 Magnetic field dependence and ground state ordering of the $\nu = 1$ state. a, b** $\rho_{xx}$ as a function of doping and $B$ at $T = 0.1$ K for $D > 0$ in device D1 (**a**) and at $T = 0.05$ K for $D < 0$ in device D2 (**b**). Cuts of $\rho_{xx}(B)$ are shown at $\nu = 1$ in the panels to the right of each map, at positions indicated by the black dashed lines in the main panels. Note that $\nu$ is the fast sweeping axis in these measurements, and as a consequence, the measured $\rho_{xx}$ is substantially smaller than its true value for the insulating states; Supplementary Fig. 8a shows a more faithful measurement of $\rho_{xx}$ at $\nu = 1$ in device D1. **c** Calculated energy of valley magnon formation in tMBG with $\theta = 1.16°$ at selected values of $\epsilon$. The inset shows the profile of the intervalley exciton, $F(\mathbf{k})$, as a function of crystal momentum in the first Brillouin zone, with the magnitude represented by a log color scale. $F(k) \neq F(-k)$, indicating that the Q-IVC state breaks TRS. $k_{x,y}$ have units $1/a_M$, where $a_M$ is the moiré lattice constant. **d** Bloch sphere representation of the VP and IVC states. VP states point towards the north (south) pole for K (K') polarization, as indicated by the red (purple) arrows. IVC states (blue arrow) point along any direction in the $x$–$y$ plane (shaded).

breaking TRS (Fig. 2c inset). The IVC order additionally prefers to carry a non-zero crystal momentum **Q**, because the valley magnon dispersion has a minimum at $\mathbf{Q} \neq 0$ (Fig. 2c). We, therefore, denote this as a "Q-IVC" state, in order to distinguish from previously considered IVC orders with $\mathbf{Q} = 0$. In combination with the large Berry curvature anticipated at the band edge, this Q-IVC order is expected to result in a trivial insulating state at integer $\nu$ when gapped, but an AHE upon weakly doping the band into a metallic state. Although a full theoretical analysis is beyond the scope of this work, we find the Q-IVC state to be a plausible ground state order that is consistent with all of our observations near $\nu = 1$ for both signs of $D$ (see Methods for a discussion of alternative plausible ground states).

Determining the ground state order for the correlated metallic states at $D < 0$ is more challenging, as transport measurements cannot directly probe the topology of the state in absence of a gap. As we do not observe any insulating behavior for the $D < 0$ correlated states in device D1, the AHE we observe at $\nu = 1$ and 3 are consistent with either an ungapped VP state or an ungapped IVC state with finite angular momentum. The combination of larger $W$ and a smaller predicted $C_\nu = 1$ naively favors IVC ordering even more strongly for the $D < 0$ band compared with the $D > 0$ band, which is thought to have $C_\nu = 2$ (Supplementary Notes 2–3). However, future work will be necessary to unambiguously identify these ground state orders.

**High-field correlated Chern insulator states**. The application of a large magnetic field serves to further modify the competition between different correlated ground states, both by tuning the orbital and spin Zeeman energies and by transforming the low energy bands into a recursive series of Hofstadter minibands. Figure 3 shows Landau fan diagrams in device D1 for both $\rho_{xx}$ and $\rho_{xy}$ at different values of $D > 0$ (leftmost and middle columns, respectively). The schematics (rightmost column) denote the

well-developed gapped states observed in each map. All are anticipated within a Hofstadter butterfly picture, in which gapped states follow trajectories described by the Diophantine equation, $\nu = tn_\phi + s$, where $t, s \in \mathbb{Z}$, and $n_\phi = \Phi/\Phi_0$ is the normalized magnetic flux. We refer to gapped states using the notation $(t, s)$, where $t$ corresponds to the Chern number of the state and $s$ corresponds to the number of electrons bound to each moiré unit cell. By convention, $(|t| > 0, s = 0)$ states are referred to as "integer quantum Hall states", whereas $(|t| > 0, s > 0)$ are "Chern insulators"[26]. States with different $s$ are distinguished by color in the schematics, whereas trivial insulating states $(t = 0)$ are depicted in black irrespective of their corresponding $s$.

We observe numerous similarities between the gapped states at filling factors $0 \leq \nu < 2$ and $2 \leq \nu \leq 4$. Although far from exact, there is an approximate mapping of our observed states between $\nu$ and $\nu + 2$, especially for maps in which we observe a robust correlated insulating state at $\nu = 2$ (i.e., Fig. 3d–l). The state at $\nu = 2$ is thought to be spin-polarized at $B = 0$[14], suggesting that interactions split the fourfold degenerate conduction band into two sets of spin-polarized but valley-unpolarized bands over a wide range of $D > 0$. The remaining valley degeneracy can also be spontaneously broken, however, this often requires the assistance of a finite $B$ depending on $\nu$ and $D$ for our studied range of twist angles. Symmetry-broken states only persist to $B = 0$ around $\nu = 1$ for a small range of $D$, and we do not observe $B = 0$ symmetry breaking at $\nu = 3$ within our studied range of $\theta$. However, we find that numerous Chern insulator states emerge spontaneously at finite $B$ over a wide range of $D$ with both $s = 1$ and 3. These states are expected to be especially strong given the finite $C_\nu = 2$ of the band, consistent with our observations and very likely indicative of a full flavor polarization of the $\nu = 1$ and 3 states at high field. In this context, the reentrant insulating behavior observed over a small range of $D$ at $\nu = 1$ (Fig. 3d, g), along with the strong $(t > 0, 1)$ Chern insulators at high field, is consistent with a phase transition from a Q-IVC state at the low field to a VP state at high field.

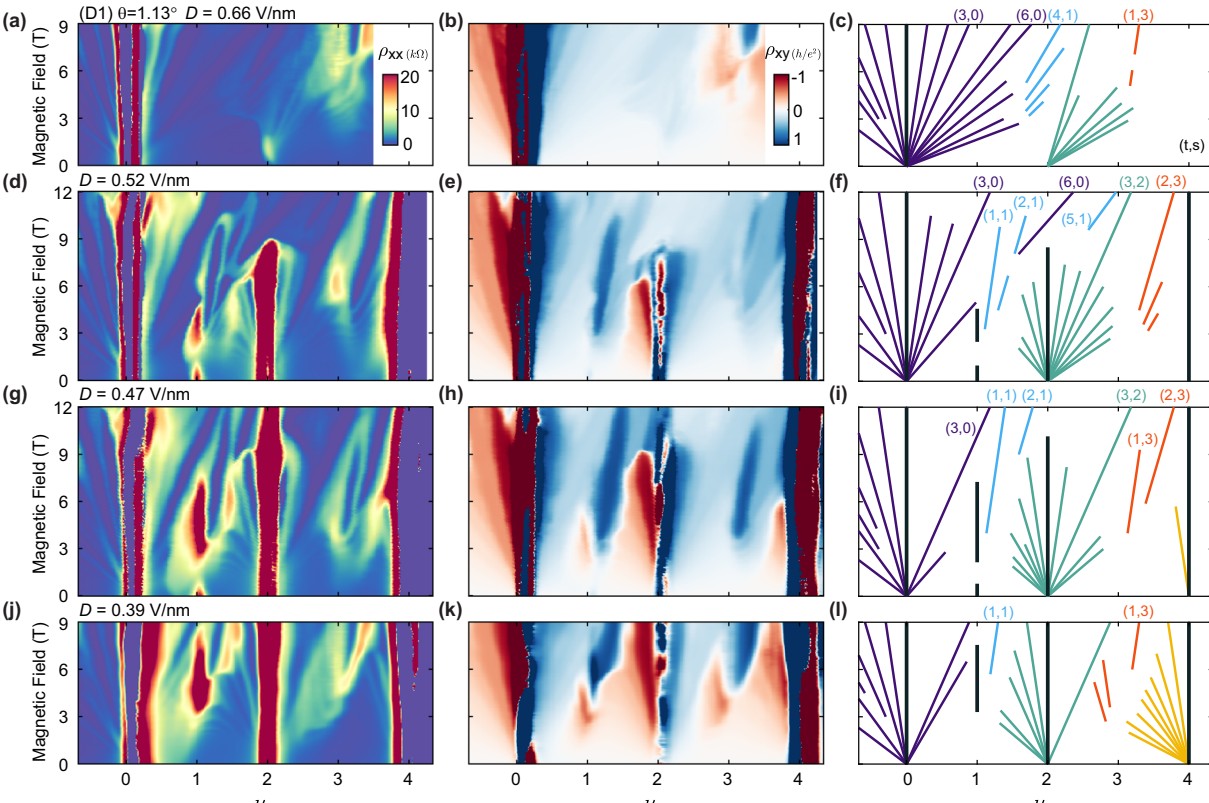

**Fig. 3 Landau fan diagrams and spontaneous flavor polarization at the high field for D > 0.** Landau fan diagrams in device D1 at (**a**, **c**) $D = 0.66$ V/nm, (**d**, **f**) $D = 0.52$ V/nm, (**g**, **i**) $D = 0.47$ V/nm, and (**j**, **l**) $D = 0.39$ V/nm, all acquired below $T = 0.3$ K. The leftmost column shows $\rho_{xx}$, the central column shows $\rho_{xy}$, and the rightmost column schematically denotes the strongest observed gapped states. In the schematic, purple lines correspond to states tracing to $\nu = 0$, blue to $\nu = 1$, green to $\nu = 2$, orange to $\nu = 3$, and yellow to $\nu = 4$. The vertical black lines denote topologically trivial insulating states. Selected states are labeled by their respective $(t, s)$ indices.

More generally, we find that the states we observe in Fig. 3 do not follow a simple progression with $\nu$ and $B$, reflecting the rich competition between the gapped single-particle Hofstadter subbands and the tendency towards spontaneous flavor polarization into a subset of these bands (see Methods for additional discussion). This is similar to recent observations in tBLG[27–32], however, here we observe a more complicated sequence of symmetry breaking in which various flavor-polarized and unpolarized ground states closely compete. For example, in Fig. 3d–f, the main-sequence quantum Hall state emanating from the charge neutrality point, $(6, 0)$, clearly intercedes both the $(2, 1)$ and $(0, 2)$ states and closes those gaps at high field. In particular, the $(2, 1)$ gap closes and then reopens at a higher field following this interruption. In addition, a $(5, 1)$ state is observed at filling factors $\nu > 2$ for $B \gtrsim 9$ T. Tuning $D$ further tips the balance in the competition between these states, highlighting their near degeneracy.

**First-order phase transitions at high field.** For $D < 0$, we observe different manifestations of field-driven competitions between correlated ground states. Figure 4a, b show Landau fan diagrams for $\rho_{xx}$ and $\rho_{xy}$ at $D = -0.37$ V/nm in device D2, with an associated schematic of the observed gapped states shown in Fig. 4c. Consistent with prior observations[14], we see the emergence of correlated insulating states at all integer $\nu$ at finite $B$. However, the improved quality of this device reveals a number of previously obscured features of these states. We observe abrupt transitions from metallic to insulating states at $\nu = 2$ and $3$ above a critical magnetic field, $B_c$, as shown in Fig. 4d. We additionally observe hysteresis at high field upon sweeping $B$ back and forth across this

phase transition at $\nu = 2$ (as well as very weak hysteresis signatures at $\nu = 3.04$), indicative of a first-order phase transition. The hysteresis additionally extends to band fillings well away from integer $\nu$. For example, Fig. 4e shows $\rho_{yx}$ at various $\nu > 3$, in which we see a hysteretic transition to a state that is nearly quantized to $h/e^2$ above $B_c$. Additional measurements in device D1 show a $(1, 3)$ Chern insulator that persists nearly to $B = 0$ (see Supplementary Fig. 3e–g), suggesting that the high-field state is likely fully flavor-polarized owing to the anticipated $C_\nu = 1$ of the band. The hysteresis we observe in Fig. 4e persists to values of $\nu$ for which the gapped states at $B < B_c$ carry $s = 4$ (see Fig. 4c), indicating a first-order topological phase transition between the flavor-unpolarized Fermi surface near full band filling and the (presumably) fully flavor-polarized Fermi surface associated with $\nu = 3$.

## Discussion

Overall, our results reveal the richness of the correlated phase diagram of tMBG, which can be tuned sensitively with the combination of $\nu$, $D$, $B$, and $\theta$. Our observation of a potential Q-IVC state at odd integer $\nu$ appears to cede to a VP state for $D > 0$ in devices with slightly larger twist angles[15], whereas flavor-unpolarized states are observed in devices with slightly smaller twist angles[14]. This suggests that the ground state order may be controlled sensitively by $U/W$, in which sequential transitions from VP to Q-IVC to flavor-unpolarized states are driven by an increasing $W$ with reducing $\theta$. Magnetic field further tunes the close competition between flavor-unpolarized ground states and numerous flavor-polarized states, resulting in abrupt and occasionally hysteretic topological phase transitions at high fields.

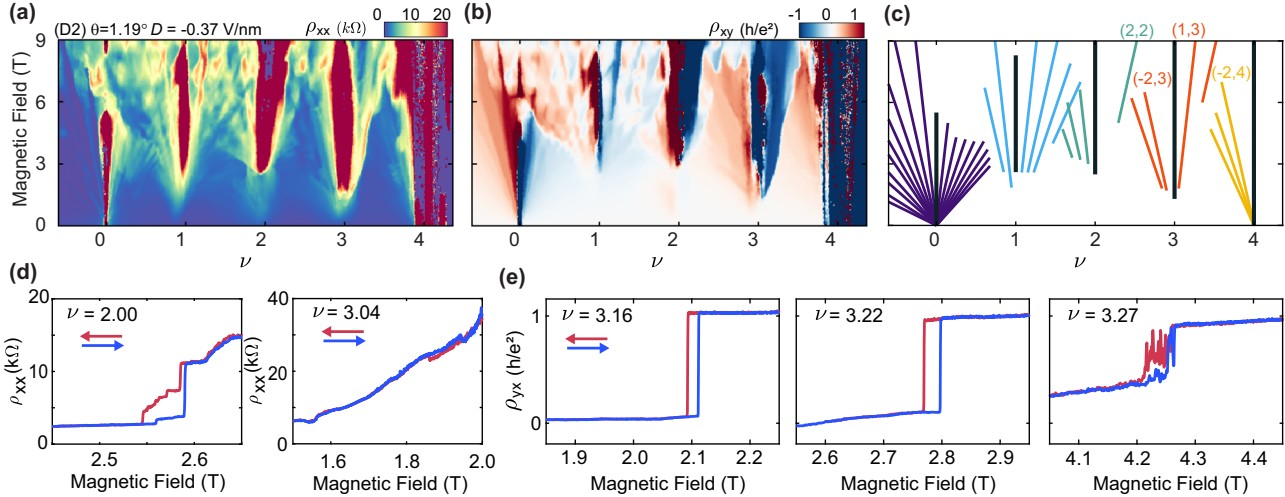

**Fig. 4 First-order orbital phase transitions at the high field for _D_ < 0. a, b** Landau fan diagram of $\rho_{xx}$ (**a**) and $\rho_{xy}$ (**b**) for device D2 at $D = -0.37$ V/nm, acquired at $T = 0.1$ K. **c** Schematic representation of the observed states, following the convention established in Fig. 3. A subset of the most robust gapped states we observe are labeled by their respective $(t, s)$ indices. States at all integer $t$ are observed for $s = 0$ and 1, indicating full degeneracy lifting. **d** $\rho_{xx}$ acquired as $B$ is swept back and forth at $\nu = 2.00$ (left) and $\nu = 3.04$ (right). **e** $\rho_{yx}$ acquired as $B$ is swept back and forth at $\nu = 3.16$ (left), $\nu = 3.22$ (middle), and $\nu = 3.27$ (right). Data in (**d**, **e**) are acquired at $T = 0.1$ K.

## Methods

**Device fabrication.** tMBG devices were fabricated using the "cut-and-stack" method[33,34], in which exfoliated graphene flakes with connected monolayer and bilayer regions are isolated using an atomic force microscope tip, and then stacked atop one another at the desired twist angle. Samples were assembled using standard dry-transfer techniques with a polycarbonate/polydimethyl siloxane stamp[35]. All tMBG devices are encapsulated in flakes of BN and graphite, and then transferred onto a Si/SiO₂ wafer. The temperature was kept below 180 °C during device fabrication to preserve the intended twist angle. Standard electron beam lithography, CHF₃/O₂ plasma etching, and metal deposition techniques (Cr/Au) were used to define the complete stack into Hall bar geometry[35].

**Transport measurements.** Transport measurements were performed in a Bluefors dilution refrigerator with heavy low-temperature electronic filtering and were conducted in a four-terminal geometry with a.c. current excitation of 1–10 nA using standard lock-in techniques at a frequency of 13.3 Hz. In some cases, a gate voltage was applied to the Si gate in order to dope the region of the graphene contacts overhanging the graphite back gate to a high charge carrier density and reduce the contact resistance. $n$ and $D$ could be tuned independently with a combination of the top and bottom graphite gate voltages through the relations $n = (V_t C_t + V_b C_b)/e$ and $D = (V_t C_t - V_b C_b)/2\epsilon_0$, where $C_t$ and $C_b$ are the top and bottom gate capacitance, $V_t$ and $V_b$ are the top and bottom gate voltage, and $\epsilon_0$ is the vacuum permittivity.

**Twist angle determination.** The twist angle $\theta$ is first determined from the values of $n$ at which the insulating states at full band filling ($\nu = \pm 4$) appear, following $n = 8\theta^2/\sqrt{3}a^2$, where $a = 0.246$ nm is the lattice constant of graphene. It is then confirmed by fitting the high-field gapped states to a Wannier diagram anticipated from the Hofstadter butterfly spectrum. The filling factor is defined as $\nu = \sqrt{3}\lambda^2 n/2$, where $\lambda$ is the period of the moiré.

**Theoretical modeling of the $\nu = 1$ ground state.** We calculate the band structure of tMBG using a standard continuum model, taking $\theta = 1.16°$ as a value intermediate to the twist angles of our measuring devices. We project the Hamiltonian to include only the conduction band and calculate the energy of a valley wave excitation assuming a fully spin-VP ground state at $\nu = 1$. We analyze the symmetry of the resulting IVC order depending on the value of $C_v$, and find that in general the profile of the intervalley exciton, $F(k)$, carries total vorticity of $2C_v$, and consequentially is not constant in momentum space for $C_v > 0$. In certain cases, finite vorticity can arise even with $C_v = 0$. This Q-IVC state is degenerated with its time-reversed partner, forming a superposition that preserves the overall TRS. However, if this degeneracy is spontaneously lifted, the resulting ground state breaks TRS. This state is also relevant at $\nu = 3$, however, we do not observe a gap at that filling in any of the devices reported here, precluding a direct experimental comparison between the Q-IVC and VP states. Detailed Hartree-Fock calculations can more faithfully assess the competition between the VP state and the various IVC orders, however, they are complicated by uncertainties in band structure parameters and details of the relevant interactions in tMBG, and are beyond the

scope of this work. Full details of our calculations can be found in Supplementary Notes 1–3.

**Additional candidate ground states at $\nu = 1$.** We discuss an additional form of Q-IVC order with associated density-wave formation consistent with our results in Supplementary Note 3. Alternatively, a more subtle origin of the behavior of the states at $\nu = 1$ is that the ground state order changes upon doping. In this scenario, the ground state at and very near $\nu = 1$ is an IVC that preserves TRS. Such an IVC state also trivializes the Chern number, resulting in a trivial insulating state when gapped. A phase transition to a fully flavor-polarized state arises upon doping away from $\nu = 1$, leading to an AHE in the metallic states owing to the inherent TRS breaking and finite $C$. Theoretically, this phase transition is expected to be first order. As we do not observe any obvious signatures of hysteresis with doping, our results are naively inconsistent with this scenario. However, it remains possible that this phase transition is smeared by various forms of disorder in the sample, including twist disorder. For this reason, we are not able to unambiguously distinguish between this scenario and the case of the TRS-breaking Q-IVC order detailed in the main text. Although we believe it to be unlikely to achieve a first-order phase transition between an IVC and VP state with a small amount of doping, a more detailed theoretical analysis is necessary to assess its feasibility. Another possibility is that interaction-induced mixing with remote bands renormalizes the valley Chern number to 0. In this case, remnant Berry curvate at the band edge could give rise to an AHE neighboring the trivial correlated insulator state, even in a VP state. However, we believe this to be highly unlikely owing both to the emergence of correlated Chern insulating states at small $B$ (see further discussion in the next section), and due to the absence of such a renormalization in our band structure calculations. Finally, we note that we have not performed an exhaustive search for all possible ground states at $\nu = 1$. Although there may be other states that are consistent with trivial insulating behavior at integer $\nu$ and an AHE upon doping, they are likely to be more exotic than the various IVC orders considered here.

**Determination of the valley Chern number.** Our calculated band structure has $C_v = 2$ (1) for the $D > 0$ ($D < 0$) conduction band (Supplementary Fig. 1). With full valley polarization at odd integer $\nu$, the gapped state at $B = 0$ has $C = C_v$[15]. However, in the apparent absence of gapped VP states at $B = 0$ in our devices, direct confirmation of the value of $C_v$ is no longer possible. Although we observe symmetry-broken states with apparent VP at large $B$, the magnetic field additionally transforms the conduction band into a recursive sequence of Hofstadter subbands that may carry different values of $C$ than at $B = 0$. The strongest gapped states observed at the high field are not necessarily constrained to a single value of $C$, but may in principle also depend dynamically on the doping since their trajectories differ according to the Středa formula. For example, we observe comparably robust $(1, 1)$ and $(2, 1)$ states in Fig. 3d–f, and comparable $(1, 3)$ and $(2, 3)$ states in Fig. 3g–i. The abrupt emergence of these gapped states with $B$ provides strong evidence that they are driven by spontaneous flavor polarization, and the separation of states with different $t$ at high field permits multiple such correlated Chern insulators to coexist at a given $B$. Because we only observe these states at a relatively high field for $D > 0$, we are unable to directly verify the anticipated $C_v = 2$ for the $B = 0$ conduction band, although their existence implies that $C_v > 0$.

Generically, the $D < 0$ states are similarly problematic, however in one instance we observe a robust (1, 3) state that emerges at relatively small $B$ in device D1 (Supplementary Fig. 3e–g). Although inconclusive without a corresponding QAH state at $B = 0$, it is suggestive that the $D < 0$ band has $C_\nu = 1$ as anticipated.

**Relation to $\nu = 1$ and 3 states in tDBG.** The correlated phase diagram of tDBG exhibits many qualitative similarities with that of tMBG at $D > 0$[14], with a notable exception that $B = 0$ Chern insulator states are observed in tMBG[14,15] but not in tDBG[36–40]. In particular, tDBG states at $\nu = 1$ and 3 are typically absent at $B = 0$, or only weakly insulating at low temperatures in devices with $\theta \approx 1.23°–1.30°$[38,40]. Supplementary Fig. 9 shows transport measurements of a tDBG device with $\theta = 1.30°$, acquired at $T = 50$ mK (note that this is the same as device D3 from ref. [40]). We observe correlated insulating states and surrounding "halo" features at $\nu = 1$ and 3. The lowest moiré conduction band of tDBG is also expected to have $C_\nu = 2$[25], therefore the observation of trivial insulating states at $\nu = 1$ and 3 appears to be inconsistent with ground states that are both spin- and VP. Although this behavior is reminiscent of our observations for the $\nu = 1$ state in our tMBG devices, we do not observe any signatures of the AHE in any of our measurements of $\rho_{xy}$ at or nearby these states in tDBG (Supplementary Fig. 9b, c). These observations could also be plausibly explained by some form of IVC order at $\nu = 1$ and 3. As discussed above, at least two different types of IVC states that retain TRS are possible: one with an s-wave intervalley exciton profile, and another comprising a superposition of time-reversed pairs of Q-IVC states with higher angular momentum. The preserved TRS of these states is necessary to explain the absence of the AHE upon doping. Additional work will be necessary to further interrogate the potential connections between these states and those we observe in tMBG.

## Data availability

All other data that support the plots within this paper and other findings of this study are available from the corresponding author upon reasonable request. Source data are provided with this paper.

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

## Acknowledgements

We thank Shaowen Chen, Cory Dean, Andrea Young, Ashvin Vishwanath, and David Cobden for helpful discussions. Technical support for the dilution refrigerator was provided by A. Manna. This work was supported by NSF MRSEC 1719797 and the Army Research Office under Grant Number W911NF-20-1-0211. X.X. acknowledges support from the Boeing Distinguished Professorship in Physics. X.X. and M.Y. acknowledge support from the State of Washington funded Clean Energy Institute. This work made use of a dilution refrigerator system which was provided by NSF DMR-1725221. Y.H.L. acknowledges the support of the China Scholarship Council. K.W. and T.T. acknowledge support from the Elemental Strategy Initiative conducted by the MEXT, Japan, Grant Number JPMXP0112101001, JSPS KAKENHI Grant Number JP20H00354 and the CREST (JPMJCR15F3), JST.

## Author contributions

M.H. and Y.L. fabricated the devices. M.H. performed the measurements, with assistance from Y.L. and Z.F. Y.-H.Z. performed the theoretical calculations. K.W. and T.T. grew the BN crystals. M.H., X.X., and M.Y. analyzed the data and wrote the paper with input from all authors.

## Competing interests

The authors declare no competing interests.
