## [Peer Review File · Nature Communications]

REVIEWER COMMENTS

Reviewer #1 (Remarks to the Author):

The present authors studied twisted monolayer-bilayer graphene (tMBG) systematically and found that there are a number of states hysteretic in magnetic field. In particular, the behaviour at $\nu=1$ seems to indicate an IVC state rather than a valley-polarized state. Overall, I find the paper very pleasant to read, and the data appears to be of exquisite quality. All references are properly cited, and the authors claims are well supported by the experimental data. For these reasons, I recommend its publications on Nature Communications. I only have a few comments which I wish the authors to address:

1. In Fig. 1, many of the ρ_{xy} traces do not change sign across field reversal. Can the authors give a comment on this?
2. In the caption of Fig. 2 the authors mentioned that ρ_{xx} is much less than its "true" value due to the way of scanning (ν as the fast axis). However, I don't see how that is relevant in this case, as in Fig. S8 there is no hysteresis (no AHE) in B at this filling, and I imagine there is no significant hysteresis in ν either. I do notice that the resistivity drops significantly even at 50mT. Could the authors explain better about this part?
3. Regarding the IVC-VP transition at $\nu=1$, the authors mentioned that there should be a first-order transition, but experimentally there does not seem to be any hysteresis. Any idea why?

Reviewer #2 (Remarks to the Author):

In this work, the authors performed magneto-transport measurements on twisted mono-bilayer graphene (tMBG) at finite displacement fields. They observed, depending on the sign of the displacement field, both correlated insulating states and "resistive" metallic states at integer filling factors of the low energy conduction bands. While at certain filling factor ranges near $\nu=1$ and 3, anomalous Hall effects were observed, no signatures of Chern insulators were seen, contrasting to some previous reports and the expectation from a valley polarized ground state. Through the numerical study of the excited states above the valley polarized state, they argue that the actual ground state could instead be an intervalley coherent state which can explain the absence of Chern

insulator behavior. They also showed features indicating a first-order transition from metallic state to correlated insulating state upon increasing magnetic field.

In general, this is a timely work comprising of solid experimental efforts. However, it is not convincing that it carries significant enough new results or insights in understanding the system of tMBG. The absence of Chern insulating behavior, which is not new in tMBG, is the main findings in this paper. The author provides a possible argument based on IVC state, which, in my opinion, does not stand out against other possibilities, and the numerical study it based on is oversimplified. Below I provide some detailed comments.

1. Unlike twisted bilayer graphene (tBLG), tMBG does not exhibit magic angle behavior. Therefore, even though the authors stress that their devices have slightly different twist angles compared to previous reports, the physics is similar for angles larger than the critical angle of topological transition discussed in Ref[19].

2. In Fig.1, the hysteresis plots of longitudinal resistivity always display a peak or plateau which onsets/disappear right at the coercive field. Is there a well-understood reason for this behavior? The peak/plateau seems to coincide with the steps in Hall resistivity in Fig 1.d and 1.h. Does the appearance of the plateau relate to the flipping of orbital magnetization domains?

3. The authors assumed that the Chern number of the quasiparticle bands is either $|C|=2$ at positive displacement field or $|C|=1$ at negative displacement field. This might be true in a non-interacting picture; however, when interactions are considered, there could well be mixing between conduction and valence bands. As shown in Ref.[19], this mixing can lead to bands with $C=0$ at small displacement fields. Therefore, it is plausible that interaction-induced mixing can also lead to quasiparticle bands with $C=0$ and finite berry curvatures at the point of avoided crossing. If so, it can also explain the absence of Chern insulator behavior.

4. The numerical method used for calculating excited states have some critical drawbacks. It involves projection only onto low energy conduction bands. The separation between conduction and valence bands is only about 10 meV or less at large displacement fields. As mentioned above, there could well be coupling to valence band due to Coulomb interaction which leads to qualitative differences in the quasiparticle states. The author did not mention how the valley polarized ground state is constructed. If it is from non-interacting bands, there could be a significant difference to an exact valley polarized solution. Both two factors will affect the excited spectrum and the occurrence of instability.

5. For negative displacement fields, the low energy electronic states resemble that of tBLG. In the case of tBLG, at filling factor $\nu=1$, which corresponds to filling one conduction electron per u.c., the ground state is also found to have valley coherence [see, e.g., arXiv:2009.13530] but with a finite Chern number $|C|=1$. What is the difference that leads to an IVC state in this work with $C=0$ (since the authors indicated that the IVC state is also applicable for $D<0$)? Moreover, the spectrum of the excited state differs significantly from that of tBLG as well: the bandwidth of the valley flip mode is 1-2 order of magnitude of that of tBLG.

In summary, the physics presented in this work does not convince me that it warrants publication in a Nature journal.

Reviewer #4 (Remarks to the Author):

This paper presents a detailed transport study of flat bands in twisted monolayer-bilayer graphene (tMBG) devices. What distinguishes this paper from previous tMBG studies is the exploration of twist angles intermediate to those of Ref 14 and 15. Their primary claim is the observation of a new correlated state near filling factor $\nu = 1$ that behaves differently than the states observed at other twist angles in tMBG. In the current work, the authors observe an insulating state at $\nu=1$ for displacement fields $D > 0$, but the state appears to be a topologically trivial insulator. This is concluded from the lack of an anomalous Hall effect (AHE) at $\nu=1$ and no shift of the feature in carrier density with a perpendicular magnetic field. This is in contrast to previous works at higher twist angles (ref 15), where a Chern insulator exhibiting a quantum anomalous Hall effect is observed at $\nu = 1$. Surprisingly, in the current work, they find that doping away from this trivial insulator at $\nu=1$ results in a non-zero AHE indicating broken time reversal symmetry. Applying high magnetic fields also shows signs of the recovery of a Chern insulator state. Taken all together it seems that at $\nu=1$ there is an insulating state with broken TRS, but it's not the Chern insulator state that would be expected in the case of singular valley polarization.

The dataset is detailed and high-quality, with a thoughtful and well-written analysis. This work will be valuable to the large community of researchers studying correlated states in twisted heterostructures and for graphene-based twisted systems in particular. I highly recommend it for publication in Nature Communications after the following small point is addressed:

On page 3 the authors say:

“Generically, the IVC state will retain TRS since it is a superposition of states at valleys K and K' , trivializing

the overall Chern number. However, our analysis of the IVC order parameter reveals that it prefers to carry finite angular momentum owing to the non-zero C_v of the constituent bands (Fig. 2c inset).”

It is not clear what Fig 2C inset is supposed to explain and how it supports the preceding sentence. Fig 2 inset is described as the magnitude of the IVC state, how does the magnitude plot in the BZ help us see that the state has net angular momentum? In general, the paper would benefit from a more intuitive explanation for how the IVC state has broken time reversal symmetry.

Reviewer #1 (Remarks to the Author):

The present authors studied twisted monolayer-bilayer graphene (tMBG) systematically and found that there are a number of states hysteretic in magnetic field. In particular, the behaviour at $\nu=1$ seems to indicate an IVC state rather than a valley-polarized state. Overall, I find the paper very pleasant to read, and the data appears to be of exquisite quality. All references are properly cited, and the authors claims are well supported by the experimental data. For these reasons, I recommend its publications on Nature Communications. I only have a few comments which I wish the authors to address:

We thank the reviewer for the positive assessment of our work, and recommendation of publication in Nature Communications.

1. In Fig. 1, many of the ρ_{xy} traces do not change sign across field reversal. Can the authors give a comment on this?

The ρ_{xy} traces plotted in Fig. 1 (as well as in Figs. S4, S6, S8) are raw data, and have not been antisymmetrized with magnetic field. Slight imperfections in the Hall bar geometry of the device inevitably mix ρ_{yx} with a small component of ρ_{xx} . Since ρ_{xx} is often very large near integer band filling, this can lead to an offset in ρ_{yx} which prevents it from changing sign upon field reversal. We have chosen not to antisymmetrize the ρ_{yx} in order to faithfully preserve the individual domain-driven jumps in the data. **We thank the reviewer for pointing this out, and we have added a note in the figure caption to clarify.**

2. In the caption of Fig. 2 the authors mentioned that ρ_{xx} is much less than its "true" value due to the way of scanning (ν as the fast axis). However, I don't see how that is relevant in this case, as in Fig. S8 there is no hysteresis (no AHE) in B at this filling, and I imagine there is no significant hysteresis in ν either. I do notice that the resistivity drops significantly even at 50mT. Could the authors explain better about this part?

The data in Figs. 2a-b is acquired by sweeping the top and bottom gate voltages simultaneously (i.e., sweeping the filling factor ν at fixed D) while at fixed B , and repeating at various values of B . Due to the interplay of the gate sweeping rate and the lock-in time constant of 300 ms, this measurement scheme fails to faithfully capture the maximum resistance of a very insulating state that arises over only a very small range of ν . In particular, this results in an underestimate of the value of ρ_{xx} in the $\nu = 1$ insulating state, even in the absence of any hysteresis. ρ_{xx} in Fig. S8a shows a more careful quantitative measurement of the insulating state resistivity, acquired by fixing the gate voltages and instead sweeping the magnetic field, and indeed it is much larger. The reviewer is correct that there is also a notable drop in ρ_{xx} even at very low magnetic fields. The precise origin of this behavior is currently not well known, however it is qualitatively consistent with the overall tendency towards a suppression of the insulating state at $\nu = 1$ with magnetic field explored in Fig. 2. **We have added a short note discussing this feature in the caption of Fig. S8.**

3. Regarding the IVC-VP transition at $\nu=1$, the authors mentioned that there should be a first-order transition, but experimentally there does not seem to be any hysteresis. Any idea why?

We thank the reviewer for pointing this out. We do not currently have a full understanding of the apparent transition between these two phases. Although the simplest theory predicts a first-order phase transition, there are additional possibilities including an intermediate metallic phase. Furthermore, the theory considers tuning U/W , whereas our measurements actually tune magnetic field; these may not be identical tuning knobs. Additionally, domains in the sample arising owing to twist angle disorder may smear the transition. For now, this remains an open question for future study.

Reviewer #2 (Remarks to the Author):

In this work, the authors performed magneto-transport measurements on twisted mono-bilayer graphene (tMBG) at finite displacement fields. They observed, depending on the sign of the displacement field, both correlated insulating states and “resistive” metallic states at integer filling factors of the low energy conduction bands. While at certain filling factor ranges near $\nu=1$ and 3, anomalous Hall effects were observed, no signatures of Chern insulators were seen, contrasting to some previous reports and the expectation from a valley polarized ground state. Through the numerical study of the excited states above the valley polarized state, they argue that the actual ground state could instead be an intervalley coherent state which can explain the absence of Chern insulator behavior. They also showed features indicating a first-order transition from metallic state to correlated insulating state upon increasing magnetic field.

In general, this is a timely work comprising of solid experimental efforts. However, it is not convincing that it carries significant enough new results or insights in understanding the system of tMBG. The absence of Chern insulating behavior, which is not new in tMBG, is the main findings in this paper. The author provides a possible argument based on IVC state, which, in my opinion, does not stand out against other possibilities, and the numerical study it based on is oversimplified. Below I provide some detailed comments.

We thank the reviewer for their close reading of our manuscript, and acknowledgement of the timeliness of the work and high quality of the experiments.

Overall, we would like to emphasize that although the absence of Chern insulating behavior at integer filling in graphene-based moiré materials is not new, the additional observation of neighboring anomalous Hall effect upon doping is so far unique to our measurements of tMBG. Furthermore, this phenomenology is difficult to reconcile with the previous theoretical and experimental reports on tMBG. Our theoretical analysis is simply meant to identify viable and feasible ground states that can rectify these unexpected experimental observations. We acknowledge that a more detailed theoretical analysis would be extremely valuable, and hope that our results will inspire further efforts along these lines. Nevertheless, we feel that our numerical study and accompanying theoretical analysis provides important new insights into the physics of tMBG, in particular by identifying a potential new type of IVC state (renamed as “Q-IVC” in the updated manuscript for clarity) that has not been considered previously in the literature, but feasibly explains our observations. We acknowledge that other explanations remain possible, and offer discussions of potential alternatives in the Methods section (“Additional candidate ground states at $\nu = 1$.”).

1. Unlike twisted bilayer graphene (tBLG), tMBG does not exhibit magic angle behavior. Therefore, even though the authors stress that their devices have slightly different twist angles compared to previous reports, the physics is similar for angles larger than the critical angle of topological transition discussed in Ref[19].

We would like to further emphasize that one of our unique observations is that the $\nu = 1$ state in our devices appears to be a trivial insulating state, but exhibits an anomalous Hall effect upon doping. This is distinct from the behavior of tMBG devices at slightly larger or smaller twist angles, in which the $\nu = 1$ state is an incipient Chern insulator (Refs. 14 and 15). This is indeed surprising given that the bandwidth of tMBG is not predicted to vary substantially with twist angle. Our observations suggest that there may be closely competing correlated ground states in tMBG, and even small changes in twist angle (which weakly tunes the bandwidth) could tip the balance between the two. Our numerical analysis further supports this hypothesis.

2. In Fig. 1, the hysteresis plots of longitudinal resistivity always display a peak or plateau which onsets/disappear right at the coercive field. Is there a well-understood reason for this behavior? The peak/plateau seems to coincide with the steps in Hall resistivity in Fig 1.d and 1.h. Does the appearance of the plateau relate to the flipping of orbital magnetization domains?

The peak/plateau-like features in ρ_{xx} are indeed well understood and ubiquitous from previous studies of the quantum anomalous Hall effect, both in magnetically-doped topological insulators and in graphene-based moiré systems. As detailed in the seminal work of K. Yasuda, et. al, Science 358, 1311-1314 (2017), the spike in ρ_{xx} is tied to the formation of a magnetic domain wall in a field. In short, this features arises when the domain wall between states of opposite magnetization moves to a position in between the two voltage probes. In this case, ρ_{xx} can either be small or large depending on the chirality of the edge modes along the domain wall, and in particular whether the edge modes are able to equilibrate with one another.

3. The authors assumed that the Chern number of the quasiparticle bands is either $|C|=2$ at positive displacement field or $|C|=1$ at negative displacement field. This might be true in a non-interacting picture; however, when interactions are considered, there could well be mixing between conduction and valence bands. As shown in Ref.[19], this mixing can lead to bands with $C=0$ at small displacement fields. Therefore, it is plausible that interaction-induced mixing can also lead to quasiparticle bands with $C=0$ and finite berry curvatures at the point of avoided crossing. If so, it can also explain the absence of Chern insulator behavior.

We appreciate the reviewer's caution in interpreting the Chern number calculation. Our experimental results appear to rule out the possibility that band mixing leads to $C=0$, as we observe the emergence of correlated Chern insulator states at modest magnetic fields. For example, Fig. 3 shows such states emerging for $D > 0$ for B above ~ 3 T, and Fig. S3e-g shows such a state emerging for $D < 0$ for B above ~ 1 T. We discuss this further in the Methods section ("Determination of the valley Chern number."). Our band structure calculations also suggest that the band gap separating the flat bands from the remote bands is sizeable. Given this, there is little to no modification of the bands upon considering the interaction-induced mixing. Nevertheless, we agree that this is a possible – albeit highly unlikely – alternative explanation of our results. **We have added a brief discussion of this point to the Methods section**

("Additional candidate ground states at $\nu = 1$.").

4. The numerical method used for calculating excited states have some critical drawbacks. It involves projection only onto low energy conduction bands. The separation between conduction and valence bands is only about 10 meV or less at large displacement fields. As mentioned above, there could well be coupling to valence band due to Coulomb interaction which leads to qualitative differences in the quasiparticle states. The author did not mention how the valley polarized ground state is constructed. If it is from non-interacting bands, there could be a significant difference to an exact valley polarized solution. Both two factors will affect the excited spectrum and the occurrence of instability.

We have indeed included the effects from the valence band and other remote bands below the conduction band in our theoretical analysis. This is detailed in Supplementary Information Section S2. The primary effect is to renormalize the dispersion of the conduction band through the Hartree and Fock terms. As far as we know, there is no qualitative change due to these renormalization effects. We would like to additionally stress that our calculation is primarily meant to motivate feasible ground states consistent with our observation of an insulating state at $\nu = 1$ and an AHE upon doping. We have not attempted to perform a full numerical calculation, however, we hope that our work will inspire more detailed theoretical analysis of this system in the future.

5. For negative displacement fields, the low energy electronic states resemble that of tBLG. In the case of tBLG, at filling factor $\nu=1$, which corresponds to filling one conduction electron per u.c., the ground state is also found to have valley coherence [see, e.g., arXiv:2009.13530] but with a finite Chern number $|C|=1$. What is the difference that leads to an IVC state in this work with $C=0$ (since the authors indicated that the IVC state is also applicable for $D<0$)? Moreover, the spectrum of the excited state differs significantly from that of tBLG as well: the bandwidth of the valley flip mode is 1-2 order of magnitude of that of tBLG.

Dirac crossings in tBLG are protected by C_2T symmetry, thus an eight band model is required to analyze potential ground states within the flat bands. tMBG is distinct from tBLG in that its lattice breaks C_2 symmetry. As a result, a single-particle gap opens at charge neutrality in a displacement field, separating the eight bands into two groups of four. This limits the possible correlated ground states in tMBG, and naturally excludes some possibilities (e.g., a partial-IVC state) considered in arXiv:2009.13530. Although there are some qualitative similarities between tBLG and tMBG with $D < 0$, the two systems nevertheless differ in the details of their symmetry and the possible ground states that can be constructed. In particular, the IVC we propose is constructed from a condensation of the valley-flip mode, and is argued to carry a finite momentum Q . This is distinct from the $Q=0$ IVC considered for tBLG. **We have expanded our discussion of this Q-IVC state throughout the text and in Supplementary Information Section S3 to highlight its novelty.**

In summary, the physics presented in this work does not convince me that it warrants publication in a Nature journal.

Reviewer #4 (Remarks to the Author):

This paper presents a detailed transport study of flat bands in twisted monolayer-bilayer graphene (tMBG) devices. What distinguishes this paper from previous tMBG studies is the exploration of twist angles intermediate to those of Ref 14 and 15. Their primary claim is the observation of a new correlated state near filling factor $\nu = 1$ that behaves differently than the states observed at other twist angles in tMBG. In the current work, the authors observe an insulating state at $\nu=1$ for displacement fields $D > 0$, but the state appears to be a topologically trivial insulator. This is concluded from the lack of an anomalous Hall effect (AHE) at $\nu=1$ and no shift of the feature in carrier density with a perpendicular magnetic field. This is in contrast to previous works at higher twist angles (ref 15), where a Chern insulator exhibiting a quantum anomalous Hall effect is observed at $\nu = 1$. Surprisingly, in the current work, they find that doping away from this trivial insulator at $\nu=1$ results in a non-zero AHE indicating broken time reversal symmetry. Applying high magnetic fields also shows signs of the recovery of a Chern insulator state. Taken all together it seems that at $\nu=1$ there is an insulating state with broken TRS, but it's not the Chern insulator state that would be expected in the case of singular valley polarization.

The dataset is detailed and high-quality, with a thoughtful and well-written analysis. This work will be valuable to the large community of researchers studying correlated states in twisted heterostructures and for graphene-based twisted systems in particular. I highly recommend it for publication in Nature Communications after the following small point is addressed:

On page 3 the authors say:

“Generically, the IVC state will retain TRS since it is a superposition of states at valleys K and K', trivializing the overall Chern number. However, our analysis of the IVC order parameter reveals that it prefers to carry finite angular momentum owing to the non-zero C_v of the constituent bands (Fig. 2c inset).”

It is not clear what Fig 2C inset is supposed to explain and how it supports the preceding sentence. Fig 2 inset is described as the magnitude of the IVC state, how does the magnitude plot in the BZ help us see that the state has net angular momentum? In general, the paper would benefit from a more intuitive explanation for how the IVC state has broken time reversal symmetry.

We thank the reviewer for the positive assessment of our work, and recommendation of publication in Nature Communications.

We thank the reviewer for pointing out the lack of clarity in the Fig. 2c inset, and in our description of the IVC state in general. Time reversal symmetry of the IVC requires that $F(k) = F(-k)$, where $F(k)$ describes the profile of the intervalley exciton underlying the IVC order. The inset shows that this condition is violated in our model, and hence the IVC state has broken TRS. **We have added an additional sentence in the figure caption to further clarify this point. We have also added additional descriptions of the uniqueness of the proposed IVC order in the main text and in Supplementary Information Section S3.**

REVIEWERS' COMMENTS

Reviewer #1 (Remarks to the Author):

The authors have satisfactorily answered all my concerns and I thereby recommend its publication in Nature Communications.

Reviewer #2 (Remarks to the Author):

I appreciate the authors for carefully and patiently addressing all my questions. Below are my concerns in regard to the responses:

In addressing question 3, the authors pointed to Fig. 3 and Fig. S3(e-g) for ruling out the possibility of C=0 state from band mixing with higher energy bands. However, these figures only suggest that the state is indeed a trivial C=0 state since the dominant Landau fan line originating from $\nu=1$ is vertical, i.e., not dispersing with B field. They seem not directly related to whether or not the C=0 state is originated from band mixing. A (1,1) or (2,1) line could also arise in the band mixing scenario. Also, the gap size between low and higher energy bands from Fig.S1 is about $\sim 40\text{meV}$, which is on the same scale as the Coulomb interaction. So I am afraid I have to disagree with the authors' statement that the remote-band mixing is very unlikely.

Secondly, the energy separation between conduction and valence band is very close. Therefore, the valence band states should be considered as active degrees of freedom, not simply as background that causes renormalization. In other words, band mixing between conduction and valence bands should be allowed and is crucial since it can also change the Chern number.

In the collective excitation spectrum, one would expect that the interaction-induced instability becomes stronger as interaction strength is increased, i.e., as screening constant ϵ is decreased. However, Fig.2(c) is showing the opposite trend with stronger instability at larger ϵ . In general, as interaction strength is decreased (or as ϵ is increased), the collective spectrum becomes closer to simply the inter-band excitations of the quasiparticles in the ground state. The trend in Fig.2(c) is indicating that if one further increases ϵ to infinity (i.e., non-interacting

limit), one would still have an instability that is even stronger. I think this is somewhat problematic. (By the way, I think it would be helpful to show the quasiparticle dispersion of the assumed spin-valley polarized ground state on top of which the collective excitations are calculated.)

Because of the above concerns, I am still suspicious that the Q-IVC state is a favorable scenario than other possibilities. However, I agree with the authors' standpoint of inspiring more studies, so I would agree to publish this work given that the above questions are addressed.

Reviewer #3 (Remarks to the Author):

I feel the authors have sufficiently addressed the reviewers concerns. I recommend publication of the revised version in Nature Communications.

REVIEWERS' COMMENTS

Reviewer #1 (Remarks to the Author):

The authors have satisfactorily answered all my concerns and I thereby recommend its publication in Nature Communications.

We thank the reviewer for recommending publication of our work.

Reviewer #2 (Remarks to the Author):

I appreciate the authors for carefully and patiently addressing all my questions. Below are my concerns in regard to the responses:

In addressing question 3, the authors pointed to Fig. 3 and Fig. S3(e-g) for ruling out the possibility of $C=0$ state from band mixing with higher energy bands. However, these figures only suggest that the state is indeed a trivial $C=0$ state since the dominant Landau fan line originating from $\nu=1$ is vertical, i.e., not dispersing with B field. They seem not directly related to whether or not the $C=0$ state is originated from band mixing. A (1,1) or (2,1) line could also arise in the band mixing scenario. Also, the gap size between low and higher energy bands from Fig.S1 is about $\sim 40\text{meV}$, which is on the same scale as the Coulomb interaction. So I am afraid I have to disagree with the authors' statement that the remote-band mixing is very unlikely.

We apologize for the ambiguity in our prior response. The reviewer is indeed correct that vertical features seen in the referenced figures are indicative of a $C=0$ state. Critically, however, we additionally observe the emergence of clear Chern insulating states (i.e. dispersing states) under modest magnetic fields. The magnetic field is expected to favor fully spin-valley polarized states, thus the emergence of these Chern insulator states at low fields appears to be consistent with a finite valley Chern number of the bands, whereas the $C=0$ states at low field instead arise from intervalley coherence. Similar physics has been observed and modeled theoretically over the past year in numerous studies of magic-angle twisted bilayer graphene (see P. Stepanov *et al.*, arXiv:2012.15126 as one example).

We nevertheless agree that there is likely some appreciable amount of remote band mixing, although it is not obvious that this should be sufficiently strong to renormalize the valley Chern number to zero. Even if this were the case, we believe that our observations remain inconsistent with a simple spin-valley polarized state at $B=0$, given the existence of the apparent phase transition driven by a small magnetic field shown in Fig. 2a. Since the spin-valley polarized state is favored at large field, this phase transition implies there should be a *different* ground state near zero field.

Finally, we note that it is also possible to construct a similar Q-IVC state even with $C=0$. Thus, our physical picture does not rely on knowledge of the valley Chern number. **We have added a sentence to the Methods section highlighting this possibility.** Beyond this, we feel that our discussion in the Methods section already acknowledges the difficulties in unambiguously determining the valley Chern number from our experiments.

Secondly, the energy separation between conduction and valence band is very close. Therefore, the valence band states should be considered as active degrees of freedom, not simply as background that causes renormalization. In other words, band mixing between conduction and valence bands should be allowed and is crucial since it can also change the Chern number.

We thank the reviewer for pointing out this important consideration. Such a rigorous calculation would be extremely valuable. However, we believe that the dominant instability of the spin-valley polarized state is likely an IVC irrespective of whether the valley Chern number is zero or non-zero, and could therefore plausibly explain our experimental observations either way. Although we do not expect band mixing to renormalize the valley Chern number to zero, we feel that this would be best addressed in a separate, more directed theoretical effort.

In the collective excitation spectrum, one would expect that the interaction-induced instability becomes stronger as interaction strength is increased, i.e., as screening constant ϵ is decreased. However, Fig.2(c) is showing the opposite trend with stronger instability at larger ϵ . In general, as interaction strength is decreased (or as ϵ is increased), the collective spectrum becomes closer to simply the inter-band excitations of the quasiparticles in the ground state. The trend in Fig.2(c) is indicating that if one further increases ϵ to infinity (i.e., non-interacting limit), one would still have an instability that is even stronger. I think this is somewhat problematic. (By the way, I think it would be helpful to show the quasiparticle dispersion of the assumed spin-valley polarized ground state on top of which the collective excitations are calculated.)

To clarify our approach, we start in the infinite U/W limit (i.e., the strongly interacting limit), in which it is now widely agreed that the ground state is spin-valley polarized (for example, see Physical Review Letters 124, 187601 (2020)). In this limit, the isospin ferromagnetic state is robust, and the valley magnon is clearly gapped. Decreasing U/W favors an instability of the ferromagnetic state, as is reflected in our calculation in Fig. 2c. Further decreasing U/W (i.e., increasing ϵ to infinity) eventually becomes no longer meaningful once the underlying spin-valley polarized state is destroyed, and the system returns to an uncorrelated Fermi liquid. Although our calculation reflects the tendency towards an instability of the spin-polarized state upon decreasing U/W , we acknowledge that this is simply meant to highlight that this instability arises at a physically reasonable value of ϵ .

Our modeling attempts to address an extremely challenging theoretical regime in which $U/W \sim 1$. By starting in the strong-coupling limit, our calculation suggests a competing IVC state that emerges upon condensing the valley magnons as U/W is reduced. However, we acknowledge that the spin-valley polarized and IVC states may not be smoothly connected, and our analysis certainly does not prove that they are. We nevertheless believe that our calculations are appropriate to provide helpful intuition for the range of ϵ that we plot.

Because of the above concerns, I am still suspicious that the Q-IVC state is a favorable scenario than other possibilities. However, I agree with the authors' standpoint of inspiring more studies, so I would agree to publish this work given that the above questions are addressed.

We thank the reviewer again for sharing their insightful viewpoints on our analysis. To

summarize, we openly acknowledge that the Q-IVC state is not the only possibility to explain our results, although we think that it is the simplest. Whatever the underlying state, we agree with the reviewer in hoping that our work will inspire further investigation into this rich system.

Reviewer #3 (Remarks to the Author):

I feel the authors have sufficiently addressed the reviewers concerns. I recommend publication of the revised version in Nature Communications.

We thank the reviewer for recommending publication of our work.